# Effect of Tooth Agenesis on Mandibular Morphology and Position

**DOI:** 10.3390/ijerph182211876

**Published:** 2021-11-12

**Authors:** Agnieszka Jurek, Dariusz Gozdowski, Ewa Monika Czochrowska, Małgorzata Zadurska

**Affiliations:** 1Department of Orthodontics, Medical University of Warsaw, 02-091 Warsaw, Poland; ewa.czochrowska@wum.edu.pl (E.M.C.); malgorzata.zadurska@wum.edu.pl (M.Z.); 2Department of Biometry, Warsaw University of Life Science, 02-787 Warsaw, Poland; dariusz_gozdowski@sggw.edu.pl

**Keywords:** agenesis, mandible, cephalometry

## Abstract

Congenital missing teeth (OMIM #106600) is the most common dental abnormality. The aim of the study was to evaluate the effects of tooth agenesis on the total mandibular length, length of the mandibular body and alveolar process, and the mandibular anteroposterior position. The material was obtained from the Department of Orthodontics, Medical University of Warsaw. The study group included 116 patients aged 9–18 years with a congenital absence of at least two permanent tooth buds in the maxilla and/or mandible (mean: 6.2 teeth missing/patient). All patients were Caucasians: 68 (59%) females and 48 (41%) males. The control group included 115 patients without tooth agenesis matched with the age and gender of the study group. A cephalometric analysis was performed, and it was focused on assessing anteroposterior mandibular measurements. This assessment was based on 17 measurements (12 linear and 5 angular). Statistical analysis of the cephalometric measurements between the study group and the control group showed significant changes regarding selected mandibular measurements. Tooth agenesis does not affect the total length of the mandible and the length of the mandibular body, but it might reduce the length of the mandibular arch length and result in a more retrusive mandibular position.

## 1. Introduction

Tooth agenesis (OMIM #106600), i.e., congenital absence of a tooth bud, is the most common dental abnormality [1,2,3,4]. It may occur in deciduous dentition or permanent dentition. Depending on its severity, we can recognize anodontia (OMIM #206780), which is a complete absence of tooth buds (defined as aplasia if it affects both deciduous and permanent dentition), oligodontia (OMIM #604625) when at least six permanent teeth are missing, and hypodontia (OMIM #106600) with a congenital absence of less than six permanent teeth. The consequences of tooth agenesis on the facial skeleton structure have been the subject of many studies, and the results vary. Some authors have not found a significant correlation between the presence of a congenital lack of permanent tooth buds and the jaw structure, while others believe that there may be significant correlations between agenesis and the facial skeleton structure [5,6,7,8,9,10,11,12,13,14,15,16,17].

The objective of this work was to assess a relationship between the presence of permanent tooth agenesis and selected anterior-posterior measurements of the mandible. We present an evaluation of the effects of permanent tooth agenesis on the length of the body and alveolar part of the mandible.

## 2. Materials and Methods

Material for the study was obtained from the Department of Orthodontics, Medical University of Warsaw.

The inclusion criteria into the study group were as follows:Age above 9 and below 18 years;Confirmed congenital lack of at least 2 permanent tooth buds in the maxilla and/or mandible;Patients with agenesis, without loss of teeth caused by other factors (caries, injuries);Patients without a history of orthodontic treatment;Good-quality panoramic radiograph;Digital lateral cephalogram.

The exclusion criteria from the study group were as follows:Age below 9 or above 18 years;Patients with congenital malformations of the facial skeleton accompanied by congenital lack of permanent tooth buds, e.g., Down syndrome, cleft lip and palate;Patients with a history of injuries to the facial skeleton;Patients during or after orthodontic treatment;Patients with premature loss of deciduous teeth;Patients with premature loss of permanent teeth;Patients with hypo-hyperdontia or anodontia of permanent dentition.

The control group included patients with presence of all permanent tooth buds.

The inclusion criteria into the control group were as follows:Age above 9 and below 18 years;Generally healthy patients;Patients without a history of orthodontic treatment;Good-quality panoramic radiograph;Digital lateral cephalogram.

The exclusion criteria from the control group were as follows:Age below 9 or above 18 years;Congenital malformations and dysmorphism of the facial skeleton, e.g., Down syndrome, cleft lip and palate;A history of injuries to the facial skeleton;Premature loss of deciduous teeth;Premature loss of permanent teeth;Patients during or after orthodontic treatment.

The analysis included data obtained from the medical history, clinical examination, assessment of panoramic radiographs and cephalometric analysis. A proprietary cephalometric analysis was developed and it was aimed at assessing anteroposterior mandibular measurements based on 17 measurements (12 linear and 5 angular) (Table 1 and Table 2, Figure 1a,b and Figure 2a,b). Cephalometric analysis was performed using the Facad Orthodontic Tracing Software (Ilexis AB, Linköping, Sweden).

Statistical analysis of the results was performed using the t-Student test and Statistica software (version 13.1) (TIBCO Software Inc. (2017), Hillview, CA, USA). The level of statistical significance has been set at *p* = 0.05. Co-Id was a variable taken into account in simple size determination of the studied groups. Based on previous studies, it was assumed that the expected difference between the mean values of Co-Id will be approximately 2 (90 vs. 92), while the standard deviation will be approximately 5. Assuming the power of the test is 80% and the confidence level is 95%, the required minimum sample size will be for each group 109 observations (patients).

Twenty-nine cephalograms were used for measurement once again after 2 months. Values of Dahlberg errors and relative Dahlberg errors indicate very low differences between the first and the second measurements for selected patients. It confirms that methods of the measurements are reliable and give repeatable results in subsequent measurements for all variables included in the study.

## 3. Results

The study group included 116 patients: 68 (59%) females and 48 (41%) males, Caucasian, aged 9–18 years. The mean age of patients was 13.4 years with a standard deviation of 6.3 years. The control group included 115 patients: 73 (63.5%) females and 42 (36.5%) males, Caucasian, aged 9–18 years, in whom the presence of all permanent tooth buds was confirmed on panoramic radiographs. The mean age in the control group was 13.6 years, with a standard deviation of 4.9 years. Two to twenty-two congenitally missing teeth were in the study group with the mean of 6.2 teeth per patient. In the anterior segments, 267 (37%) teeth were missing, and in the posterior segments—453 (63%). In the maxilla, there were 354 (49.2%) missing teeth, and 366 (50.8%) in the mandible. The distribution between the right and left sides was symmetrical, with 360 teeth missing on each side. In 33 (28.4%) patients in the study group, two teeth were missing. Agenesis of 6 teeth was present in 14 (12.0%) patients, of 5 teeth in 12 (10.3%) patients, of 4 in 10 (8.6%) patients, and of 8 teeth in 9 (7.8%) patients (Table 3). The number of patients with hypodontia was 60 (51.7%), and with oligodontia—56 (48.3%).

The most frequently missing teeth included mandibular second premolars. Agenesis of the lower left second premolar was observed in 70 (60.3%) patients and of lower right second premolar in 72 (62.0%) patients, followed by maxillary lateral incisors (right lateral incisor was missing in 56 (48.3%) patients, and left one in 53 (45.7%) patients) and maxillary second premolars (tooth 15 was missing in 55 (47.4%) patients, and tooth 25 in 49 (42.3%) patients). Agenesis of maxillary central incisors and mandibular first molars were the least common (Table 4).

In 18 (15.5%) patients, tooth agenesis was observed only in the maxilla and in 20 (17.2%) only in the mandible. In the study group, 65 patients had symmetrical agenesis in the maxilla and 66 in the mandible, corresponding to 56.0% and 56.9%, respectively. Bilateral absence of tooth buds in both upper and lower dental arches occurred in 32 (27.5%) patients, while unilateral agenesis in 13 (11.2%) patients. If teeth were missing only in the anterior segment, they always included maxillary lateral incisors and mandibular incisor. Absence of central incisors and canines in the maxilla and mandibular canines was present only in cases of severe agenesis. In the posterior segments, a greater discrepancy was noted, and both isolated agenesis of premolars and isolated agenesis of molars were observed.

Statistical analysis of the cephalometric values between the study and the control groups showed significant changes in selected measurements (Table 5). In the study group, the distance of Id, B and Pg points from the PA line was significantly smaller, which may indicate a more posterior position of these points. The distance between Id and B points and the PM line was also smaller compared to the control group. A significantly smaller width of the mandibular symphysis measured between D and Pg points and a smaller value of the B and Id mental angles were found; B and Id points were located more posteriorly in relation to the Pg point.

## 4. Discussion

There were no statistically significant differences in the measurements of the total mandibular length (Co-Pg), mandibular body length (Go-Pg) and mandibular alveolar part length (Go-B). The values of Co-Pg, Go-B and Go-Pg segment lengths were smaller in the study group than in the control group, but the difference was not statistically significant. This result indicated that in the study group, the presence of tooth agenesis did not significantly affect the total mandibular length and the length of its body. Wisth et al. measured the distance between the Pogonion point and the point corresponding to the Go point used in the study and showed that the mandibular body was shortened [5]. Similar results were obtained by Biedziak who measured the mandibular length in patients with oligodontia using the Garn method. A comparison with the norms of the average measurements in the sagittal dimension obtained from the analysis performed in the study group showed a shortening of the mandibular body and ramus and a reduced height of the mandibular body in the anterior segment [10]. Misevska et al. showed statistically significant mandibular shortening only in patients with agenesis of one to two teeth; they did not observe such changes in the group with more severe agenesis [18]. No effects of agenesis on the mandibular length were also observed by Øgaard and Krogstad, Endo et al. and Zadurska et al. in their studies [7,11,12,16,17].

Statistical analysis did not reveal any association between a congenital absence of permanent teeth and the mandibular length measured between the Co point and the largest depression on the anterior chin outline (Co-B), as well as between the Co point and the point on the anterior margin of the mandibular alveolar apex (Co-Id). In contrast, a statistically significantly decreased distance of Id and B points from the PA and PM lines was observed. The study showed no statistically significant changes in the length of the PgPg” segment, but the PgPg’ segment was statistically significantly shorter in the study group compared to the control group.

In the study, tooth agenesis did not affect the mandibular anteroposterior position (SNB and SNPg angles). Gungor and Turkkahraman also did not observe any significant differences in the SNB angle values [19]. Øgaard and Krogstad found no statistically significant changes in the anteroposterior position of the mandible [7]. On the other hand, Lisson and Scholtes showed significantly larger SNB angle and SNPg angle in the study group compared to the control group, but without statistically significant differences between the group with hypodontia and the group with oligodontia [20]. Kreczi et al. also observed a reduced SNB angle in the study compared to the standard value and in the group with mandibular agenesis and in the group with agenesis in the maxilla and mandible compared to the standard value [14]. Misewska et al. showed statistically significant retrognathic mandible (reduced SNB angle) [18]. Taju et al. obtained results demonstrating different effects of agenesis on the SNB angle value, depending on the ethnic group; they observed both a reduction and no effects of missing tooth buds on a retrusive position of the B point [21]. In the studies by Yüksel and Üçem, the mandible was more protrusive (a greater SNB angle) in the case of bilateral agenesis in the posterior segment than unilateral in the anterior and posterior segments and in the control group, and bilateral agenesis in the anterior segment than unilateral agenesis in the anterior segment. The SNGn angle was smaller in the group with bilateral agenesis in the anterior segment than in the control group and unilateral anterior agenesis and the unilateral posterior agenesis [8]. Mayama et al. found no significant difference in SNB measurement between the oligodontia group and the control group and a negative correlation between a position of the B point and the number of missing teeth; the higher the number of missing teeth, the more retrusive a position of the B point [22]. Nodal et al. obtained results showing a more prognathic mandible (a position of the Pg point relative to the nasion point) in patients with more than 13 permanent teeth missing and increasing chin prognation with an increasing number of missing teeth, while the prognathism of the mandibular alveolar part in his study decreased with an increasing number of missing teeth [6]. In her studies, Biedziak demonstrated an anterior position of the body and chin in relation to the cranial and maxillary base in patients with agenesis of four or more permanent teeth [10]. However, Kumar et al. obtained results indicating flattening of the chin in a group of patients with teeth agenesis in the maxillary anterior segment [15].

In our study, a statistically significant smaller diameter of the mandibular symphysis was observed in the study group compared to the control group. Tavajohi-Kermani et al. obtained results showing a positive correlation between the chin button thickness and missing teeth in the maxilla and in the maxilla and mandible simultaneously [9].

The analysis of mandibular angle size (ArGoGn) showed no significant change between the study and control groups. Endo et al. in their studies also observed no statistically significant differences in the gonial angle [11]. Nodal et al. observed a reduced angle, and it was statistically significant in patients who were missing at least 13 permanent tooth buds [6].

The anteroposterior position of the B and Id points relative to the Pg point was statistically significantly more retrusive. Biedziak, Sarnäs and Rune, and Nodal et al. measured the CL/ML angle, which corresponds to the mental angle Id used in this study, and also showed its reduction [6,10,23]. In contrast, Ben-Bassat and Brin found no statistically significant changes in the value of the mental angle [24].

Despite of the lack of statistically significant changes in the Co-Pg length, Co-B, Go-B and Go-Pg segments, shortening of BB′, BB″, PgPg′ and PgPg″ was observed.

## 5. Conclusions

Tooth agenesis has no effect on the total mandibular length and on the mandibular body length, but it may result in a shortening of the mandibular arch length and more retrusive position of the mandible.

## Figures and Tables

**Figure 1 ijerph-18-11876-f001:**
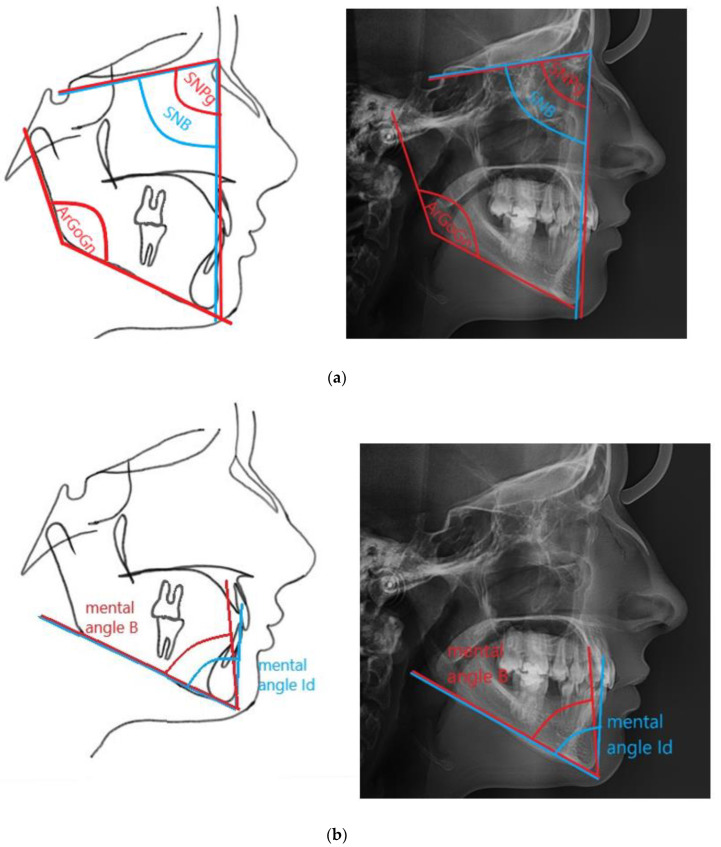
Angular measurements of the mandible used in the study. (**a**) shows mandibular angle and anteroposterior position of the mandible and the chin; (**b**) shows anteroposterior position of the B point and Id point, relative to the Pg point.

**Figure 2 ijerph-18-11876-f002:**
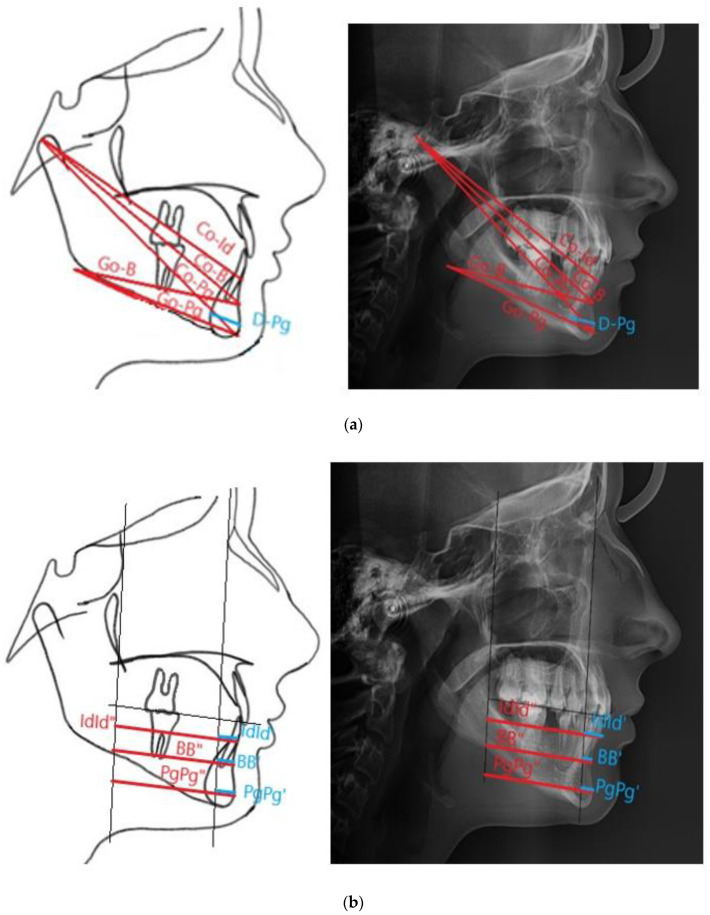
Linear measurements of the mandible used in the study. (**a**) shows the distance of selected points from the top of the condylar process and the angle of the mandible; (**b**) shows the distance of the points Id, B and Pg from the lines PA and PM.

**Table 1 ijerph-18-11876-t001:** Cephalometric points, planes and construction points used in the study.

Cephalometric Points	Cephalometric Planes	Construction Points
Ar—articulare, the point of intersection of the inferior outline of the cranial base with the posterior outline of the mandibular neck	FOP—functional occlusal plane, passing through M and pM points	B′—the point of intersection of a straight line parallel to FOP passing through the B point with the PA line
B—supramentale, the deepest point located on the anterior outline of the mandible	HL—horizontal line, perpendicular to PM and passing through SE	B″—the point of intersection of a straight line parallel to FOP passing through the B point with the PM line
Ba—basion, the most posteroinferior point of the cancellous part of the clivus, lying in the median plane	NB—a vertical line marked by N and B points, defining anteroposterior position of mandible	FMS’—projection of the FMS point on the HL line
Co—condylion, the most posterior and superior point of the mandibular head	NPg—a vertical line determined by N and Pg points, defining an anteroposterior position of the chin	Go—gonion, the point of intersection of the line tangential to the lower margin of the mandible and tangential to the posterior outline of the mandibular ramus
D—in our study, the most posterior point on the posterior outline of the mandibular symphysis	ML—a mandibular base line, the line indicated by Me and tgo1 points	Id′—the point of intersection of a straight line parallel to FOP passing through the Id point with the PA line
FMS—frontomaxillary nasal suture, the most superior point of the suture at the anterior connection of the nasal bone and maxilla	MT—a line of the posterior margin of the mandibular ramus, indicated by Ar and tgo2 points	Id″—the point of intersection of a straight line parallel to FOP passing through the Id point with the PM line
Gn—gnathion, the most anterior and inferior point on the lower outline of the chin	PA—a straight line parallel to the PM passing through FMS	Pg′—the point of intersection of a straight line parallel to FOP passing through the Pg point with the PA line
Id—the point on margin of the mandibular alveolar process between central incisors, or on top of the alveolar process in the midline in case of mandibular incisor agenesis	PM—a vertical line passing through the SE and Pm points	Pg″—the point of intersection of a straight line parallel to FOP passing through the Pg point with the PM line
M—the most posterior and inferior contact point of the last erupted upper molar with the opposite tooth	PgB—a line passing through Pg and B points	Sd′—the point of intersection of a straight line parallel to FOP passing through the Sd point with the PA line
N—nasion, the most anterior point of the frontonasal suture	PgId—the line passing through Pg and Id points	Sd″—the point of intersection of a straight line parallel to FOP passing through the Sd point with the PM line
Pg—pogonion, the most anterior point on the outline of the chin		
pM—a contact point of the most mesially located premolars, deciduous molars or canines in the case of agenesis of all deciduous premolars and molars		
Pm—pterygomaxillare, the point of intersection of the outline of the posterior maxillary contour or the anterior contour of the pterygopalatine fossa with the outline of the hard palate		
S—sella, the point located in the geometric center of the sella turcica, in the medial plane		
SE—sphenoethmoidal, the point located at the intersection of the cranial base and the greater wings of the sphenoid bone		
tgo1—the point on the inferior margin of the mandibular angle at the point connecting a line tangential to the lower margin of the mandibular body		
tgo2—the point on the posterior margin of the mandibular angle at the point connecting a line tangential to the posterior margin of the mandibular body		

**Table 2 ijerph-18-11876-t002:** The linear and angular measurements used in this study.

Linear Measurements	
BB′	The distance between B and B′, in the authors’ study defined as the length of the anterior mandibular segment (the authors’ measurement)
BB″	The distance between B and B″
Co-B	The distance measured between Co and B points (the authors’ measurement)
Co-Id	The distance measured between Co and Id points (the authors’ measurement)
Co-Pg	The total mandibular length, the distance measured between Co and Pg points
D-Pg	The cross-sectional diameter of the mandibular symphysis, the distance between D and Pg points
Go-B	The length of the mandibular alveolar part, distances between Go and B points (the authors’ measurement)
Go-Pg	The mandibular body length, the distance between Go and Pg points
IdId′	The distance between Id and Id′, defined as the anterior length of the mandibular alveolar part (the authors’ measurement)
IdId″	The distance between Id and Id″
PgPg′	The distance between Pg and Pg′, defined as the length of the anterior mandibular part at the Pg height (the authors’ measurement)
PgPg″	The distance between Pg and Pg″
Angular Measurements	
ArGoGn	The mandibular angle, the angle betweenArGo andGoGn lines
Mental angle B	The angle between the ML mandibular plane and PgB line, determining the anteroposterior position of the B point, relative to the Pg point (the authors’ measurement)
Mental angle Id	The angle between the ML mandibular plane and the PgId line, determining the anteroposterior position of the Id point, relative to the Pg point
SNB	The angle determined by the SN and NB lines that determines the anteroposterior position of the mandible
SNPg	The angle determined by the SN and NPg lines that determines the anteroposterior position of the chin

**Table 3 ijerph-18-11876-t003:** Number of missing teeth in the study group.

Number of Missing Teeth	Number of Patients	Percent Value
2	33	28.7%
3	5	4%
4	10	9%
5	12	10.4%
6	14	12.2%
7	5	4%
8	9	8%
9	4	3.4%
10	6	5%
11	4	3.4%
12	2	2%
13	2	2%
14	4	3.4%
15	2	2%
16	2	2%
18	1	0.9%
22	1	0.9%

**Table 4 ijerph-18-11876-t004:** The distribution of missing teeth in the study group.

Tooth (Maxilla)	Number of Missing Teeth in the Study Group	% of Missing Teeth in the Whole Group	Tooth (Mandible)	Number of Missing Teeth in the Whole Group	% of Missing Teeth in the Study Group
11	2	0.28%	31	37	5.14%
12	56	7.8%	32	16	2.2%
13	13	1.8%	33	10	1.4%
14	30	4.16%	34	24	3.3%
15	55	7.64%	35	70	9.7%
16	7	0.97%	36	2	0.28%
17	18	2.5%	37	28	3.9%
21	4	0.55%	41	39	5.42%
22	53	7.9%	42	13	1.8%
23	13	1.8%	43	11	1.53%
24	30	4.16%	44	17	2.36%
25	49	6.8%	45	72	10%
26	6	0.83%	46	2	0.28%
27	18	2.5%	47	25	3.47%

**Table 5 ijerph-18-11876-t005:** Results of statistical analysis between the study and the control groups.

Measurement	Mean ± Standard Deviation		*p*	Lower Quartile (Q1) and Upper Quartile (Q3)	
	Study Group	Control Group		Study Group	Control Group
Co-Pg	103.17 ± 7.37	104.01 ± 6.43	0.354	98.9–107.5	100.5–109.5
Co-B	93.23 ± 6.37	93.73 ± 5.55	0.522	90.0–97.6	91.0–98.5
Co-Id	90.22 ± 7.95	91.71 ± 5.48	0.098	87.7–94.9	88.9–96.2
SNB	79.11 ± 4.52	78.79 ± 3.54	0.550	97.9–104.2	98.5–103.2
SNPg	80.57 ± 4.78	79.86 ± 3.63	0.206	96.5–102.8	97.1–102.6
ArGoGn	134.4 ± 6.41	133.84 ± 6.09	0.494	131.2–138.0	128.9–137.1
D-Pg	13.86 ± 2.15	14.64 ± 2.26	0.007 *	12.7–15.1	13.1–15.7
Go-B	63.29 ± 5.59	63.91 ± 5.15	0.382	49.3–55.2	50.8–56.2
Go-Pg	68.35 ± 6.01	68.68 ± 5.02	0.652	64.0–72.2	66.0–73.0
IdId″	43.07 ± 4.51	45.48 ± 4.27	0.000 *	40.2–46.2	42.9–48.3
IdId′	8.96 ± 5.59	12.59 ± 5.28	0.000 *	4.7–13.7	9.7–16.2
BB′	7.96 ± 5.64	10.61 ± 5.45	0.000 *	3.8–12.5	6.9–14.6
BB″	41.35 ± 5.13	43.08 ± 4.83	0.009 *	38.5–44.5	39.9–46.6
PgPg′	10.61 ± 6.91	13.03 ± 6.72	0.007 *	5.3–16.3	8.4–17.5
PgPg″	44.03 ± 6.4	45.38 ± 6.35	0.108	40.0–47.9	41.3–50.6
Mental angle B	60.48 ± 18.4	64.1 ± 5.9	0.045 *	56.8–66.9	60.8–68.0
Mental angle Id	68.11 ± 18.12	73.35 ± 5.22	0.003 *	65.2–73.9	69.8–76.5

* statistically significant values are marked with the asterisk (*p* < 0.05).

## Data Availability

Raw data are available from the corresponding author on reasonable request.

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
