# Peer review of "Effect of Tooth Agenesis on Mandibular Morphology and Position"

_ijerph, 2021, doi:10.3390/ijerph182211876_

Round 1
Reviewer 1 Report
Dear Authors,
I would like to congratulate you on your work. The topic is really interesting and the work is well structured and explained.
However, Introduction needs to be expanded, Discussion and Conclusions do not give a clear explanation on how the results can be a useful addition and contribution to literature and eventually to the clinical practice or future researches.
You will find my other comments by clicking on the highlighted parts of the attached manuscript. Please, address each one of them.

Author Response
Thank you very much for revision.
I introduced the suggested changes in red in the attached file.
In response to your comments:
Point 1: Why age range above 9 and below 18 years was chosen?
Response 1: According to the literature all tooth buds can be assessed after the age of 9. We included growing patients in the study.
Point 2: How congenital lack of at least 2 permanent tooth buds without loss of teeth caused by caries of injures were assessed and verified?
Response 2: These two criteria were assessed and verified based on panoramic radiograph, clinical examination and data obtained from the medical history.
Point 3. Table 2 is a bit redundant.
Response 3. I leave it for the editorial decision.
Point 3: What does the study add to literature? How is this study original and brings something new compared to other works? What are its limitations?
Response 3: The knowledge of the effects of agenesis of permanent dentition on the length parameters of the mandible allows as to predict the consequences and to select an appropriate treatment method. This is another study in the literature that assesses effect of the congenital missing teeth on the mandible parameters. This can be used for comparison with others. The study covered a relatively large group.

Reviewer 2 Report
Great article with comprehensive measurement. However my minor comment is on the line 58. Can author explain why they exclude Hypodontia case while we are assessing patient with missing several number of teeth?
Author Response
Thank you very much for your review.
In response to your comments:
Point 1 Can author explain why they exclude Hypodontia case while we are assessing patient with missing several number of teeth?
Response 2. We excluded patients who had both hypodontia and hyperodontia in permanent dentition.
Reviewer 3 Report
The manuscript "Effect of tooth agenesis on mandibular morphology and position" is a comprehensive scientific research with high valuable data. The hypothesis that patients with teeth agenesis could present different mandibular morphology is interesting, but the results of the study didn't confirm this relation. It is hard to explain why the mandibular position was more retrusive compared with control group, but due to genetic etiology factors of teeth agenesis it could be suggest that these group of patients present different growth direction. I would like to propose further examination of the material in the next study with assessment of maxillary morphology.
The manuscript present high valuable scientific data of the craniofacial morphology patients with agenesia. Cephalometric measurements of the mandible are comprehensive with wide description both in tables and in figures. The strength of the study is accuracy in construction of the sample: well designed inclusion and exclusion criteria and calculation of the number of participants. and bias assessment. Weakness is lack of clinical implementation in Discussion part. It may be interesting for practitioners but not obligatory in scientific paper.
Author Response
Thank you very much for your review.
We will make further examination based on this material.